# Spatial genetic diversity and populational differentiation of *Ternstroemia sylvatica* (Ericales: Pentaphylacaceae) in eastern Mexico

**Hernán Alvarado-Sizzo**[1,2], **Isolda Luna-Vega**[2], **Edna Arévalo-Marín**[3], **Othón Alcántara-Ayala**[2], **Gerardo Rivas**[4]*

1 Jardín Etnobotánico y Museo de la Medicina Tradicional y Herbolaria - Centro INAH Morelos, Secretaría de Ciencia, Humanidades, Tecnología e Innovación, Cuernavaca, Morelos, Mexico, 2 Facultad de Ciencias, Departamento de Biología Evolutiva, Universidad Nacional Autónoma de México, Ciudad de México, Mexico, 3 Centro de Investigaciones Interdisciplinarias en Ciencias y Humanidades - CEIICH, Universidad Nacional Autónoma de México, Ciudad de México, Mexico, 4 Facultad de Ciencias, Departamento de Biología Comparada, Universidad Nacional Autónoma de México, Ciudad de México, Mexico

* gerardorivas@ciencias.unam.mx

## Abstract

*Ternstroemia sylvatica* inhabits several temperate and tropical montane forests in eastern Mexico. Its current discontinuous distribution results from both natural and anthropogenic fragmentation. We assessed the genetic diversity and population differentiation of *T. sylvatica* across its distribution range using 18 microsatellite markers. We sampled 366 individuals from 16 populations, analyzing genetic diversity (*He*) and population structure via STRUCTURE and Discriminant Analysis of Principal Components (DAPC). Our results revealed high genetic differentiation ($F_{ST} = 0.21$), with most genetic variation occurring within populations (79.50%). STRUCTURE analysis identified two major genetic clusters: a northern group, comprising the populations with the lowest genetic diversity, and a southern group with higher genetic diversity (*He* = 0.59–0.73) geographically structured into ten subgroups. Additionally, the results suggest historical fragmentation, limited gene flow among populations and inbreeding, as a heterozygote deficit is prevalent across populations. The high genetic diversity in specific populations indicates potential hybridization with other sympatric *Ternstroemia* species.

## Introduction

*Ternstroemia sylvatica* Schltdl. & Cham. (1830) is a shrubby or little tree species endemic to eastern Mexico (Fig 1), where it is distributed mainly in the Sierra Madre Oriental (SMOr), eastern Trans-Mexican Volcanic Belt (TMVB), and with some populations in the northern part of the Sierra Madre del Sur (SMS). It inhabits the tropical montane cloud forest (TMCF) and temperate pine and pine-oak forests at elevations

**Data availability statement:** All relevant data are within the manuscript.

**Funding:** G.R. received funding from PAPIIT-IN220621, Dirección General del Personal Académico (DGAPA), Universidad Nacional Autónoma de México (UNAM). I.L.-V. received funding from PAPIIT- IN219424, Dirección General del Personal Académico (DGAPA), Universidad Nacional Autónoma de México (UNAM). H.A.-S. received a Postdoctoral Grant 2022-2024 from Dirección General del Personal Académico (DGAPA), Universidad Nacional Autónoma de México (UNAM).

**Competing interests:** The authors have declared that no competing interests exist.

between 1300–1450 m asl. In Mexico, the TMCF has a discontinuous distribution, primarily associated with narrow altitude ranges and certain slope expositions (shady and moist slopes) [1]. This distribution pattern results from historical climate changes such as the Pleistocene glacial cycles and the complex landscape heterogeneity of the SMOr [2], in the case of *T. sylvatica*, its distribution area consists of discontinuous patches driven by biogeographic features, the most conspicuous being "La Gran Sierra Plegada" (SGSP) that separates the northernmost population of El Cielo, Tamaulipas, from the populations of Sierra Gorda in northern Querétaro State (ca. 180 km in straight line) and the Moctezuma River basin (MB), a region of about 70 km broad that separates the populations of Sierra Gorda from the southern part of the SMOr. Historical and ongoing land-use change has fragmented forest areas in eastern Mexico, including the *T. sylvatica* distribution. These regions are considered critical or of medium priority for conservation purposes [3]. Additionally, the potential impacts of harvesting *T. sylvatica* flowers and fruits for medicinal purposes (té de tila) [4], which is quite common in the region, remain unknown. In particular, no studies have evaluated how such extraction may affect the long-term sustainability or persistence of natural populations.

Recent population genetics studies on tree and tree-like species from the TMCF and other montane humid forest species such as *Alsophila firma* [5], *Chiranthodendron pentadactylon* [6], *Dicksonia navarrensis* [7], *Liquidambar styraciflua* [8], *Magnolia* sect. *Macrophylla* [9], *Magnolia schiedeana* [10], *Magnolia tamaulipana* [11], and *Podocarpus matudae* [12] documented pronounced spatial genetic structure at regional scales and often limited gene flow among populations (e.g., 5–7, 9, 12). In addition, studies on *Magnolia tamaulipana* [11], *Oreopanax xalapensis* [13], and *Pinus chiapensis* [14] examined the effects of disturbance and demography, demonstrating a positive association between population size and genetic diversity. Wehenkel et al. [15] recently demonstrated that fragmentation resulted from logging and land-use change due to agriculture and cattle raising affects the genetic diversity and future conservation of tree species by causing habitat isolation, reduced gene flow between populations, and increased edge effects.

Montane cloud-forests trees of eastern Mexico remain genetically undercharacterized [16–18], and *Ternstroemia sylvatica* lacks a range-wide baseline on how genetic diversity is partitioned within and among populations, or whether major topographic barriers structure that variation. Here, we address this gap by sampling across the species' distribution and genotyping with a set of 18 microsatellites markers to: 1) evaluate the genetic diversity and differentiation and 2) analyze the relationship between the main biogeographic disjunctions (specifically SGSP and MB gaps) on genetic diversity and differentiation patterns.

## Materials and methods

### Tissue collection and DNA isolation

We collected young leaves in silica gel from 366 individuals of *Ternstroemia sylvatica* in 16 populations, sampling all its known distribution in TMCF from SMOr (Fig 2, Table 1). Collections were made during several field trips between 2020 and 2023

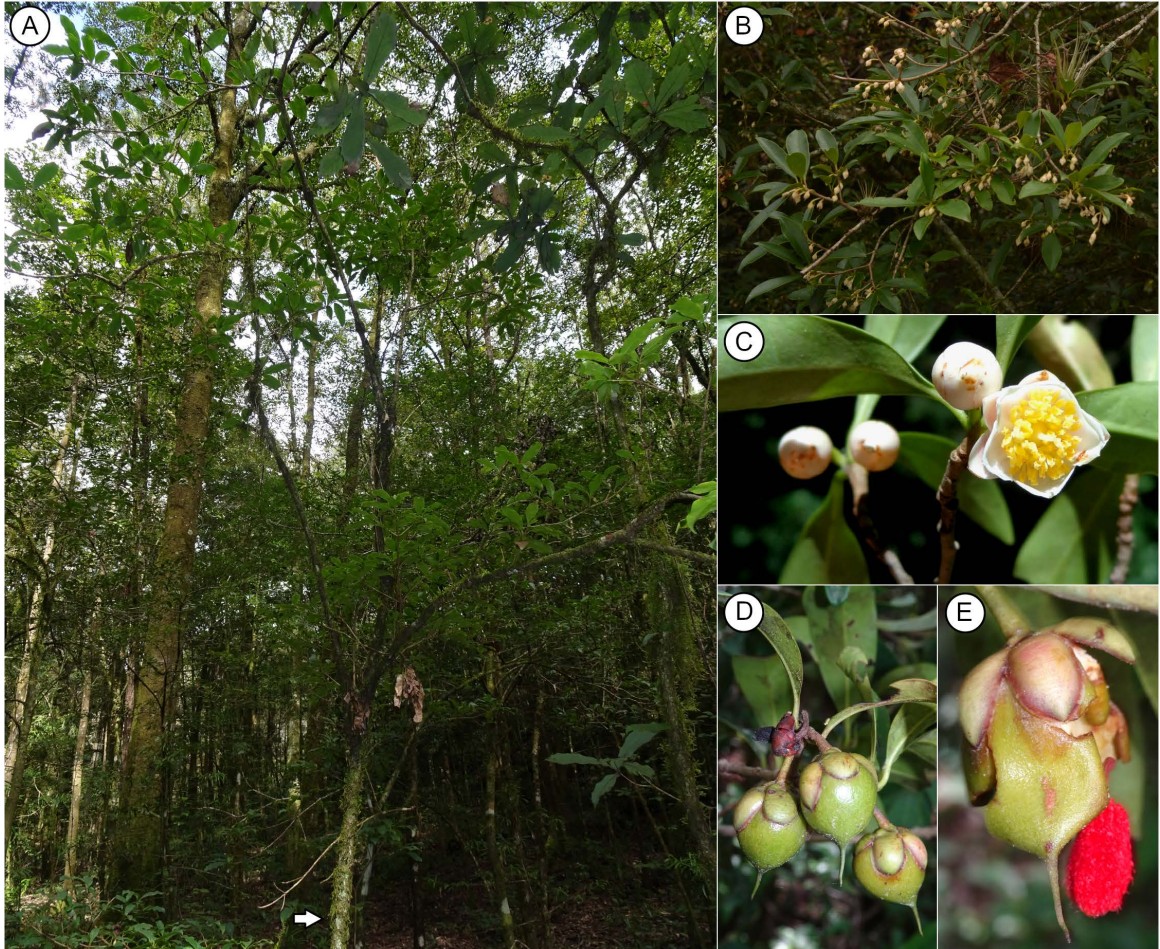

**Fig 1. Pictures of *T. sylvatica*.** (A) General view of a typical tree (foreground with arrowhead). (B) Detail of a branch showing the arrangement leaves. (C) Open flower and flower buds. (D) Fruits. (E) Open fruit showing a seed. Photo credits: (A, C, D, and E) Leccinum Jesús García-Morales, Reserva de la Biosfera del Cielo, Tamaulipas, Mexico. (B) José Rodrigo Carral Domínguez, Ixhuacán de los Reyes, Veracruz, Mexico.

(Collection permit SGPA/DGGFS/712/2924/18 issued by SEMARNAT). We ground the leaves in individual 2 mL micro-tubes using a Retsch Mill (MM400). DNA was isolated using the CTAB protocol [19].

## Genotyping

We used a subset of 18 microsatellite markers designed by Alvarado-Sizzo et al. [21] and successfully tested in *T. huas-teca* among other *Ternstroemia* species. The microsatellite markers were pooled in four multiplex PCR reactions, each containing 4 or 5 fluorescently labeled oligos. PCR reactions were assembled according to the manufacturer's instructions for the Platinum Direct PCR Universal Master Mix (Thermo-Fisher, USA). Markers were pooled in four multiplex reactions containing four or five oligo pairs. The annealing temperature was 60 °C since all the primers were designed around this value. The fragment analysis was performed in Psomagen Inc. (Maryland, USA), and genotyping was achieved using the Microsatellite plugin (v. 1.4.7) of Geneious Prime 2022 (Dotmatics, NZ). Allele scoring was performed manually using the supplementary material of Selkoe & Toonen [22] as guidelines.

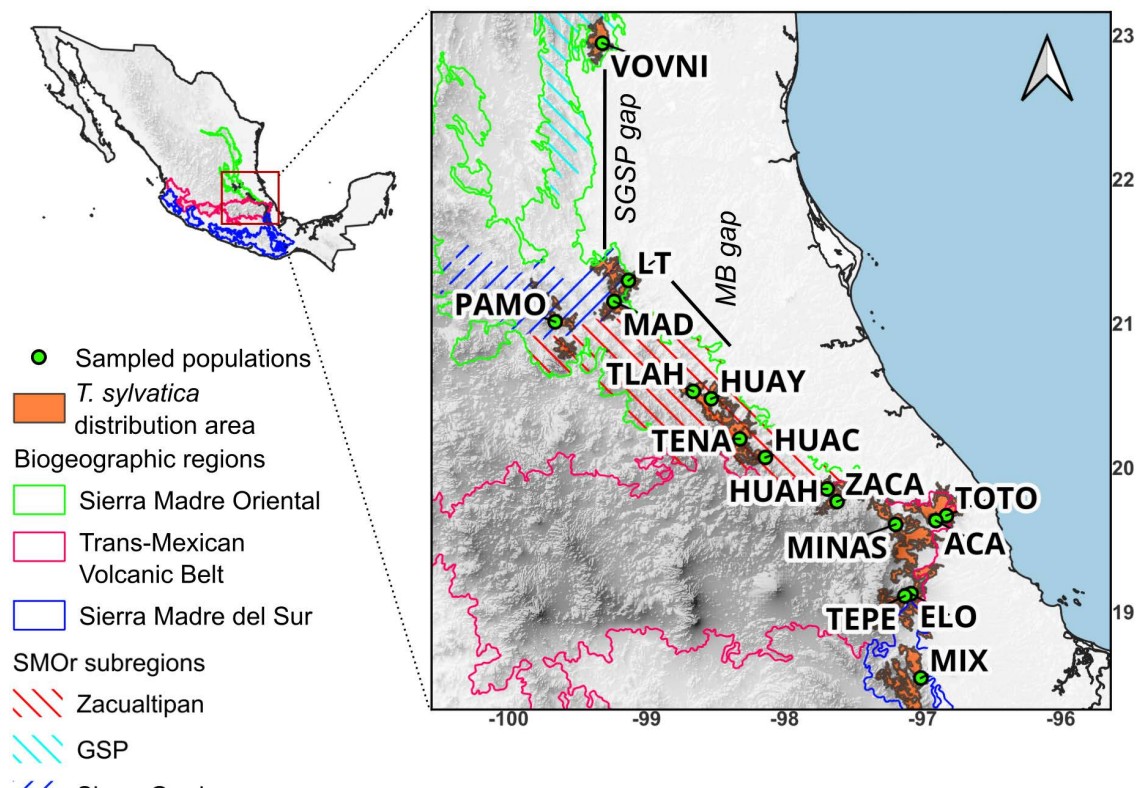

**Fig 2. Distribution of *T. sylvatica* in México and sampled populations.** *SGSP gap*: Gran Sierra Plegada Gap, *MB gap*: Moctezuma River basing gap. All Population Acronyms are defined in Table 1. Basemap from Continuo de Elevaciones Mexicano v 3.0, Instituto Nacional de Estadística y Geografía (México) [20].

**Table 1. Location of sampled populations in this study.** Coordinates are Geographic decimal using WGS84. SMOr = Sierra Madre Oriental, TMVB = TransMexican Volcanic Belt, SMS = Sierra Madre del Sur, GSP = Gran Sierra Plegada, SG = Sierra Gorda, ZAC = Zacualtipán district, - = no subregion considered, X = Longitude, Y = Latitude, N = number of sampled individuals per population.

| ID | Acronym | Population | Region | Subregion | X | Y | N |
|----|---------|-----------|--------|-----------|---|---|---|
| 1 | VOVNI | Valle del OVNI | SMOr | GSP | −99.22598949 | 23.05892904 | 28 |
| 2 | LT | La Trinidad | SMOr | SG | −99.06652681 | 21.40639949 | 24 |
| 3 | MAD | El Madroño | SMOr | SG | −99.17643963 | 21.26287174 | 29 |
| 4 | PAMO | Pinal de Amoles | SMOr | SG | −99.61481425 | 21.1286298 | 26 |
| 5 | TLAH | Tlahuelompa | SMOr | ZAC | −98.60599296 | 20.6265038 | 30 |
| 6 | HUAY | Huayacocotla | SMOr | ZAC | −98.47493905 | 20.57222183 | 24 |
| 7 | TENA | La Cruz de Tenango | SMOr | ZAC | −98.27194167 | 20.28878474 | 17 |
| 8 | HUAC | Huauchinango | SMOr | ZAC | −98.0879098 | 20.15479305 | 21 |
| 9 | HUAH | Huahuaxtla | SMOr | ZAC | −97.64128328 | 19.92395643 | 15 |
| 10 | ZACA | Zacapoaxtla | TMVB | – | −97.57201436 | 19.8340994 | 25 |
| 11 | TOTO | Totoyac | TMVB | – | −96.77481417 | 19.71161966 | 25 |
| 12 | ACA | Acatlán | TMVB | – | −96.85332233 | 19.67944193 | 20 |
| 13 | MINAS | Las Minas | TMVB | – | −97.14887068 | 19.66275942 | 20 |
| 14 | ELO | Elotepec | TMVB | – | −97.05273641 | 19.1797569 | 20 |
| 15 | TEPE | Tepehuacan | TMVB | – | −97.0994672 | 19.16478862 | 22 |
| 16 | MIX | Mixtla | SMS | – | −97.00294166 | 18.59260208 | 20 |

## Genetic analysis

Before performing the diversity and differentiation analyses, we screened the microsatellites set evaluating null alleles, Hardy-Weinberg, and linkage disequilibrium using the R packages *PopGenReport* [23], *pegas* [24], and *poppr* [25], respectively. Additionally, we built an accumulation curve to assess our locus set and the power to discriminate individual multilocus genotypes (MLG). We considered these previous analyses as the "marker set suitability check." We calculated the general genetic diversity and differentiation parameters using the R packages *adegenet* [26], *hierfstat* [27], and *poppr* [25]. We performed genetic clustering using STRUCTURE 2.3.4 [28] under the "No Admixture" model, using 1 000 000 MCMC after discarding the first 100 000 as burn-in and iterating the process ten times from $K = 2$ to $K = 16$. The most likely number of groups ($K$) was determined using Evanno's method [29] using Structure Harvester [30], and the ten iterations were summarized using CLUMPP. Also, we assessed population structure with Discriminant Analysis of Principal Components (DAPC) [31] implemented in the *adegenet* R package [26]. DAPC is model-free (no HWE/LD assumptions), and it is appropriate for microsatellite data and for organisms that may not freely recombine. Populations were not predefined. Instead, we inferred genetic clusters de novo using "find.clusters" (k-means on the PCA-transformed genotypes) and selected the number of clusters with the Bayesian Information Criterion (criterion = "diffNgroup"), which supported $K = 5$ clusters. We then performed DAPC using these groups. To minimize overfitting, the number of principal components retained in the DAPC (n.pca) was tuned by α-score optimization ("optim.a.score") after an initial fit, yielding n.pca = 29. Because DAPC with $K$ groups estimates at most $K$–1 discriminant functions, we retained n.da = 4. Posterior membership probabilities from the final DAPC were used to summarize each individual's genetic background, and ordinations were visualized with "scatter.dapc". We performed an Analysis of Molecular Variance (AMOVA) in the R Package *poppr* [25] using 999 repetitions for calculating the test of significance, considering variances between regions defined by the STRUCTURE first-level clustering "northern" (populations 1–6) and "southern" groups (populations 7–16), populations (within regions), and individuals. Since a genetic pattern emerged during the preliminary genetic analysis, we employed the geostatistical approach "inverse distance weighting" (IDW), as implemented in QGIS 3.22.4 [32], using the average *He* and *Ho* values by population. This task enabled us to visualize the genetic variation along the geographical distribution of *T. sylvatica*. Finally, since the first-level genetic structure matches the pattern of genetic diversity, we compared the *He* values for the "northern" and "southern" groups using a Student's t-test (*t.test* function) in the R Statistical Software (v4.3.1) [33].

## Results

### Marker set suitability check

We found no evidence of more than two alleles, which is inconsistent with polyploidy; therefore, we assumed our genotypes correspond to diploid organisms. The accumulation curve showed total individual MLG discrimination at eight loci, with no clones present (S1 File). This fact ensures that the number of loci used (18) is at least twice the required number to distinguish any pair of individuals within our sample. 13 loci showed a consistent HW deviation ($p \leq 0.05$) in the northern populations VOVNI, LT, MAD, PAMO, and TLAH (S1 File). However, these were retained in subsequent analyses since they behaved relatively normally in the remaining populations. We found no evidence of linkage disequilibrium as $\bar{r}_d$ (S1 File), just as reported in the original microsatellites description [20]. According to the null alleles test using the methods of Chakraborty et al. [34] and Brookfield [35], our data showed no significant evidence of null alleles. Given the good performance of the 18 microsatellite markers, we retained all of them for genetic analysis.

### Genetic diversity and differentiation

The 18 markers showed polymorphism when considering all the populations. The markers Tli162, Tli187, and Tli197 were monomorphic in at least one population. However, we decided to include them in all the analyses since they

could provide information for the remaining populations. La Trinidad (LT) population displayed the lowest observed heterozygosity ($Ho = 0.33$), while Pinal de Amoles (PAMO) showed the lowest values of expected heterozygosity ($He = 0.53$), the Huauchinango (HUAC) population held the highest $Ho$ (0.61), and the Tepehuacan (TEPE) population showed the highest $He$ (0.73). Mean $Ho$ and $He$ were 0.50 and 0.63, respectively. Genetic diversity was significantly lower in the northern group than the southern group, as indicated by $He$ ($t = -2.53$, df = 7.45, p-value = 0.038) (Fig 3). In all populations, $He$ was higher than $Ho$; the average inbreeding index ($F_{IS}$) was 0.26, ranging from 0.16 (TLAH and MINAS) to 0.42 (LT) (Table 2). The overall $F_{ST}$ was 0.21, and AMOVA showed that most of the variation (79.50%) occurred among individuals within populations, whereas 11.84% of the variation occurred among populations. The remaining 8.67% is explained when two groups of populations are considered (Table 3). Paired $F_{ST}$ analysis revealed that the lowest differentiation among populations occurred between two populations of the TMVB ($F_{ST} = 0.03$, TOTO vs ACA). Within this region, low genetic differentiation prevailed. For example, among the TOTO, ACA, MINAS, and ELO populations, $F_{ST}$ values were low to moderate (below 0.06). The highest paired differentiation was observed between the populations PAMO and TEPE ($F_{ST} = 0.23$). However, the TEPE population consistently showed moderate to high genetic differentiation compared to all the other populations ($F_{ST}$ from 0.12 to 0.23). Populations from northern SMOr (VOVNI, LT, MAD, and PAMO) showed low to moderate differentiation among them ($F_{ST} < 0.10$), but in most cases, held moderate to high differentiation ($F_{ST} > 0.10$) when compared to populations from South SMOr, TMVB, and the population from SMS (MIX) (Fig 4).

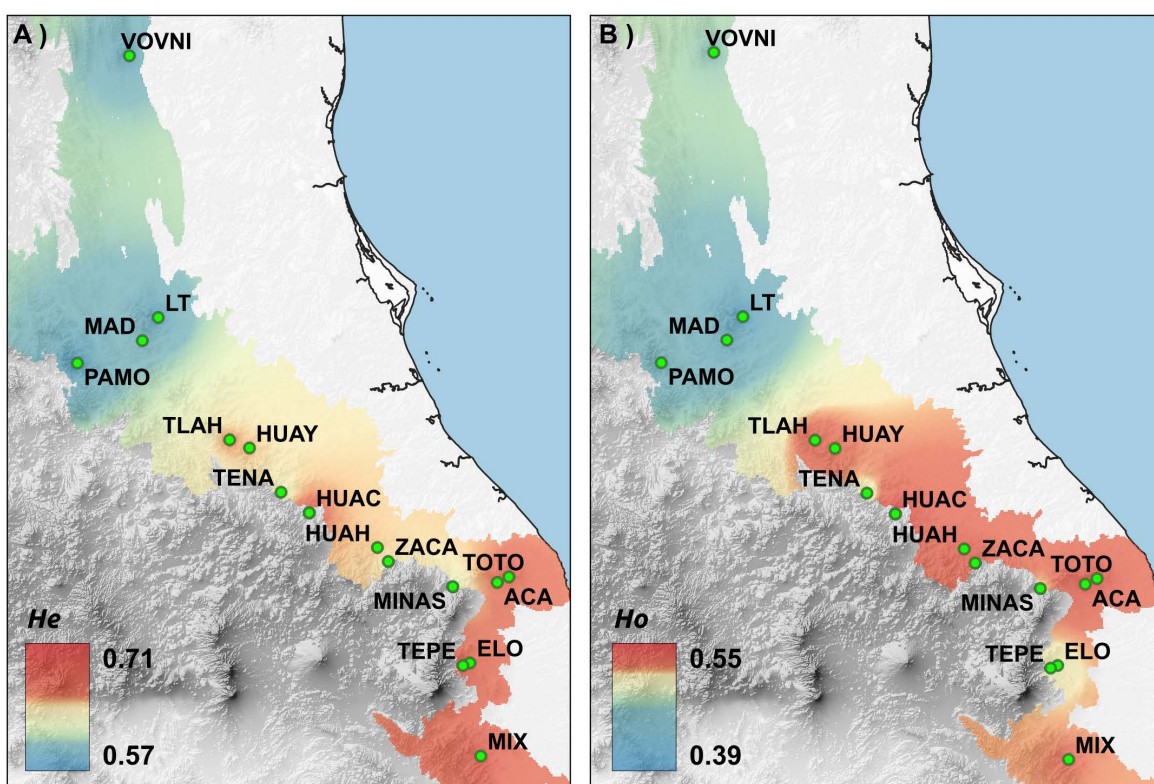

**Fig 3. IDW interpolation of A) *He* and B) *Ho* in Eastern Mexico. Basemap from Continuo de Elevaciones Mexicano v 3.0, Instituto Nacional de Estadística y Geografía (México) [20].**

**Table 2. Basic genetic diversity parameters.** n = number of individuals, *Ho* = Observed Heterozygosity, *He* = Expected Heterozygosity, Ar = Allelic Richness, $A_R$ = Rarefied Allelic Richness, $F_{IS}$ = Inbreeding index, $N_P$ = number of private alleles.

| Pop | N | *Ho* | *He* | Ar | $A_R$ | $F_{IS}$ | $N_P$ |
|---|---|---|---|---|---|---|---|
| VOVNI | 28 | 0.47 | 0.57 | 5.07 | 4.56 | 0.19 | 3 |
| LT | 24 | 0.33 | 0.54 | 4.03 | 3.21 | 0.42 | 2 |
| MAD | 29 | 0.41 | 0.60 | 5.23 | 4.20 | 0.38 | 0 |
| PAMO | 26 | 0.44 | 0.53 | 4.43 | 3.84 | 0.26 | 4 |
| TLAH | 30 | 0.56 | 0.66 | 5.46 | 4.94 | 0.16 | 1 |
| HUAY | 24 | 0.54 | 0.65 | 5.52 | 4.79 | 0.23 | 1 |
| TENA | 17 | 0.48 | 0.62 | 5.80 | 5.39 | 0.3 | 3 |
| HUAC | 21 | 0.61 | 0.69 | 7.37 | 6.57 | 0.15 | 6 |
| HUAH | 15 | 0.56 | 0.65 | 6.39 | 5.55 | 0.2 | 5 |
| ZACA | 25 | 0.53 | 0.63 | 6.39 | 5.50 | 0.25 | 6 |
| TOTO | 25 | 0.55 | 0.69 | 7.55 | 6.36 | 0.26 | 9 |
| ACA | 20 | 0.57 | 0.65 | 6.52 | 5.99 | 0.16 | 2 |
| MINAS | 20 | 0.50 | 0.59 | 5.65 | 5.05 | 0.2 | 1 |
| ELO | 20 | 0.49 | 0.63 | 5.88 | 5.06 | 0.27 | 4 |
| TEPE | 22 | 0.50 | 0.73 | 6.75 | 5.81 | 0.34 | 8 |
| MIX | 20 | 0.53 | 0.70 | 6.32 | 5.51 | 0.31 | 2 |
| **Mean** | 22.88 | 0.50 | 0.63 | 5.90 | 5.14 | 0.26 | 3.56 |

**Table 3. Analysis of Molecular Variance (AMOVA) considering variances between individuals, among populations (within regions), and regions ("northern" and "southern" groups).** SS = Sum of Squares, MS = Mean Square, Est. Var. = Estimated Variance, % Est. Var. = % Estimated Variance. *There may be variations in the second decimal place since the values are rounded to two digits.

| Source | df | SS | MS | Est. Var. | % Est. Var. | Fixation indices |
|---|---|---|---|---|---|---|
| Between Region | 1 | 173.51 | 173.51 | 0.78 | 8.67 | $F_{CT}$ = 0.09 |
| Between populations within Region | 14 | 438.18 | 31.30 | 1.06 | 11.84 | $F_{IS}$ = 0.13 |
| Between individuals | 350 | 2499.89 | 7.14 | 7.14 | 79.50 | $F_{ST}$ = 0.21 |
| **Total*** | 365 | 3111.57 | 8.53 | 8.99 | 100.00 | |

## Genetic clustering

Structure analysis showed two main genetic groups (*K* = 2) according to Evanno's method (Fig 4). One group comprises six populations from the SMOr (LT, MAD, PAMO, VOVNI, TLAH, and HUAY), referred to as the "northern group" through the study In contrast, the southern group includes three populations from southern SMOr (TENA, HUAC, and HUAH), six populations from the TMVB (ACA, ELO, MINAS, TEPE, TOTO, and ZACA), and one population from the SMS (MIX) (Figs 5 and 6).

Also, Evanno's *ΔK* indicated the presence of a substructure at *K* = 10 (Fig 4). This substructure corresponds to the split of the northern group into three genetic groups with an incipient geographic structure (Fig 7). The other eight groups are the result of the split of the southern group in the Huayacocotla group (HUAY+TLAH), Tenango group (TENA+HUAH), Chiconquiaco group (TOTO+ACA), and Xalapa group (MINAS+ELO). Four populations were identified as unique genetic clusters: Huauchinango (HUAC), Zacapoaxtla (ZACA), Tepehuacan (TEPE), and Mixtán (MIX) (Fig 7).

DAPC showed five clusters: 1) two populations (TLAH and HUAY) from northern SMOr (green in Fig 8). 2) the largest cluster (dark blue in Fig 8) comprising 10 populations from southern SMOr+TMVB (TENA, HUAC, HUAH, ZACA, TOTO, ACA, MINAS, ELO, TEPE, MIX). 3) 26 individuals from the VOVNI and MAD populations (yellow in Fig 8). 4) Most

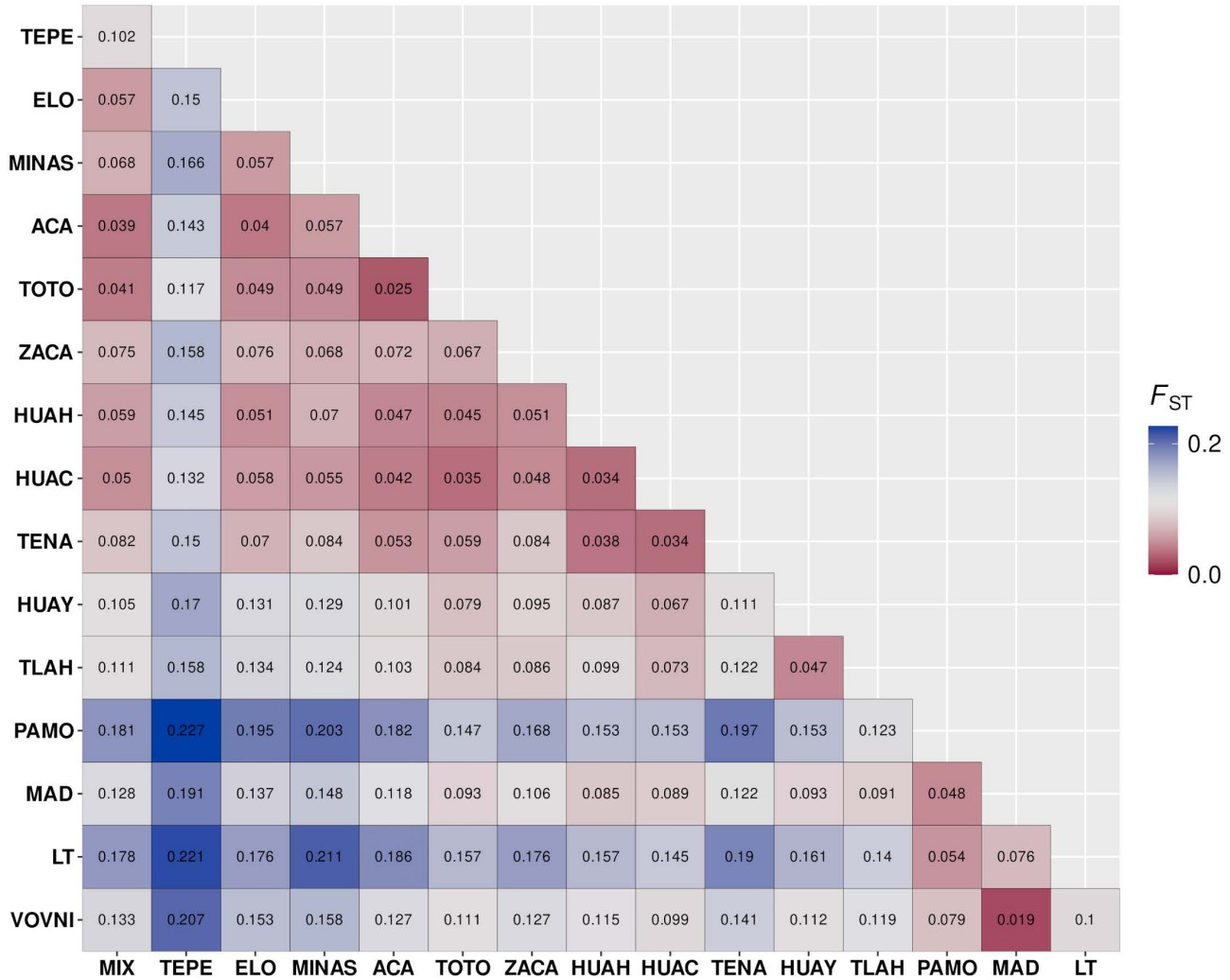

**Fig 4. Paired $F_{ST}$ among all the studied populations.**

individuals (n = 17) of the TEPE population (light blue in Fig 8). 5) four northern populations (VOVNI, LT, MAD, and PAMO) (red in Fig 8). It is noteworthy that cluster 4 diverges largely from the remaining clusters, primarily due to variation along axis 2.

## Discussion

### Genetic diversity and differentiation

The calculated genetic diversity is relatively high when compared to microsatellite-based averages (*He* = 0.56) for asterids [36] and for other threatened Pentaphylacaceae such as *Euryodendron excelsum* (*He* = 0.54) and *Ternstroemia gymnanthera* (*He* = 0.33–0.50) [37]. Genetic diversity is comparable to *T. lineata* (*He* = 0.68) which is associated with temperate vegetation, has similar fragmentation issues and was evaluated using the same microsatellites set [21].

Among the populations of *T. sylvatica*, expected heterozygosity was moderate (mean *He* = 0.63, range 0.53−073; see Table 2). However, the northern group exhibited lower genetic diversity (*He* = 0.53–0.66) compared to the southern group

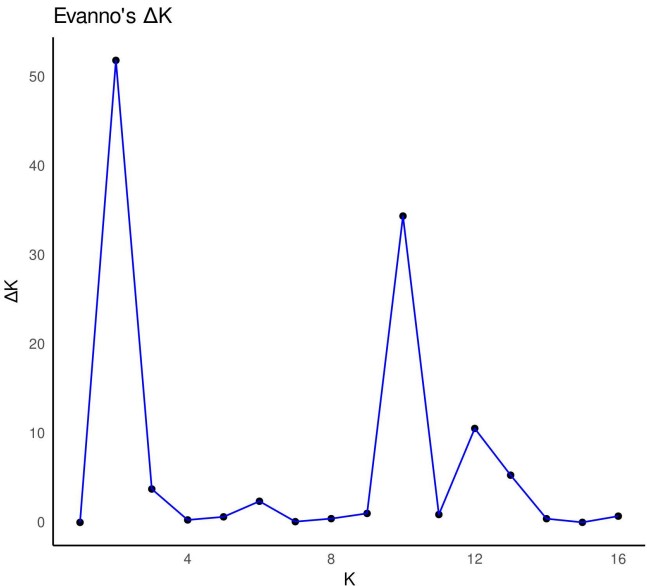

**Fig 5. Analysis of Evanno's ΔK.** *K* = 2 or first-level genetic structure, and substructure at *K* = 10.

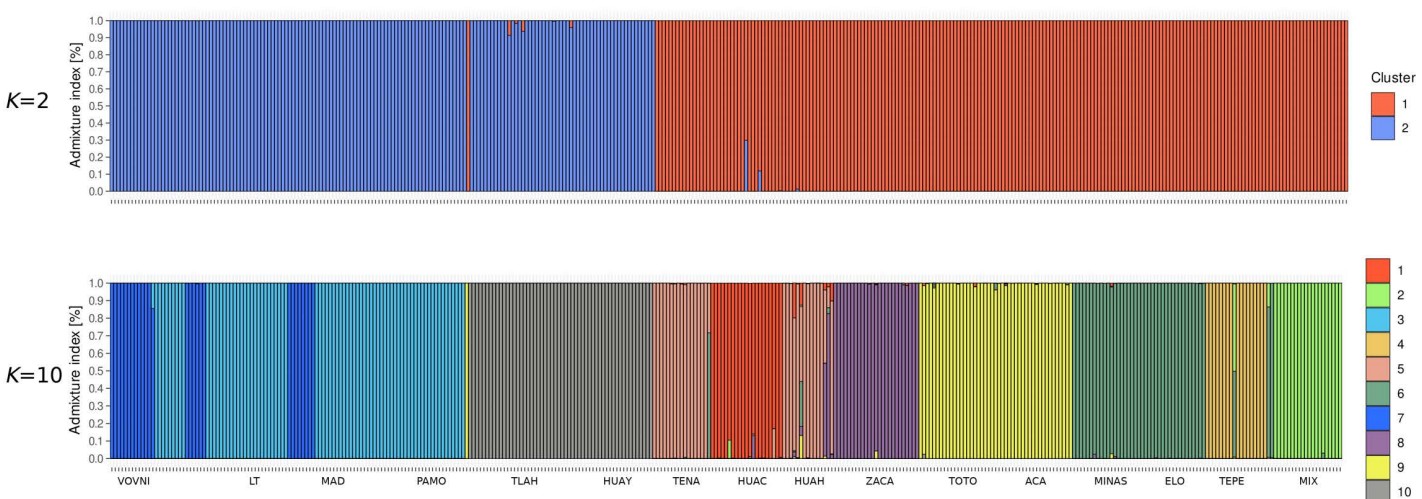

**Fig 6. STRUCTURE analysis plot.** Individual assignment to *K* = 2 and *K* = 10. Population acronyms in Table 1.

(*He* = 0.59–0.73). Specifically, the northernmost populations (VOVNI, LT, MAD, and PAMO) show *He* = 0.53–0.60, which we consider comparatively low and potentially consistent with incipient genetic erosion at the population level. Genetic diversity estimates for other TMCF trees (e.g., *Abies guatemalensis* [38], *Magnolia rzedowskiana* [39], *Pinus chiapensis* [14], and *Quercus mulleri* [40] represent a rather dissimilar sample in terms of phylogenetic relationships and life histories, making it difficult to categorize the obtained values for *T. sylvatica*. However, within our *T. sylvatica* sampling, we identified a geographic subset with reduced genetic diversity; does not allow us to determine conclusively whether the reduction is driven primarily by drift associated with small effective population sizes or by a history of isolation. Because the sampled

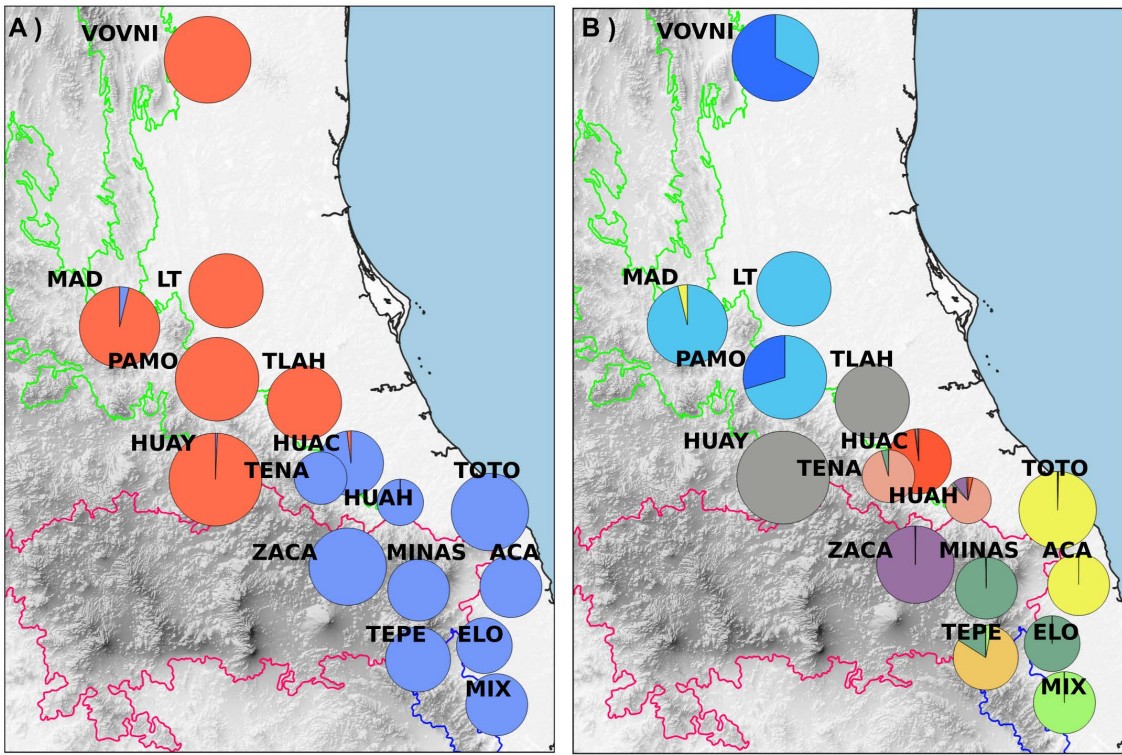

**Fig 7. Map of STRUCTURE assignment by population.** A) $K = 2$ and B) $K = 10$. Pie diameter proportional to population size. Population acronyms in [Table 1](). Basemap from Continuo de Elevaciones Mexicano v 3.0, Instituto Nacional de Estadística y Geografía (México) [20].

populations are small (≈20 adults per population, exhaustively sampled), loss or under-detection of rare alleles could contribute to the observed *He* values. We avoid labeling the identified pattern as 'genetic erosion' and describe it as reduced diversity in a regional group of the species. The disjunction between the northern and southern groups is largely concordant in terms of genetic diversity and geography: the northern group contains populations with diversity values ranging from moderate, such as HUAY and TLAH (*He* = 0.65–0.66), to low (0.53–0.60 in VOVNI, LT, MAD, and PAMO); while the southern group is characterized by populations with moderate (*He* = 0.62) to high (*He* = 0.73) genetic diversity ([Fig 3]()). Contrasting this with the biogeographic disjunctions shown in [Fig 2](), then the approximately 70 km distributional gap across the Moctezuma River basin (MB), which is part of the greater Pánuco Basin, appears to coincide with the separation of a low-diversity area, encompassing the four northernmost populations of VOVNI, LT, MAD, and PAMO from those with moderate diversity (HUAY and TLAH). This concordant disjunction (genetic and biogeographic) may be explained by historical factors such as successive fragmentations of TMCF driving to vicariance: according to the Parsimony Analysis of Endemism (PAE) of Luna-Vega et al. [1], the TMCF patches of northern Sierra Gorda and Tamaulipas form a biogeographic unit which is strongly differentiated from that of southern SMOr.

Besides the apparent geographical genetic diversity pattern (decreasing northwards), we consistently find higher values of *He* in relation to *Ho*, suggesting inbreeding as a deficit of heterozygotes, at least in the northern populations of LT ($F_{IS} = 0.42$) and MAD ($F_{IS} = 0.38$). It is noteworthy that populations showing high diversity spots (measured as *He*), such as the Huayacocotla and Tepehuacan groups, are sympatric with other *Ternstroemia* species, including *T. huasteca* and *T. acajetensis*. Also, Tepehuacan population showed high divergence in DAPC. Both high diversity spots and divergence can be taken as evidence of hybridization or introgression occurring in this region. This phenomenon is consistent with an

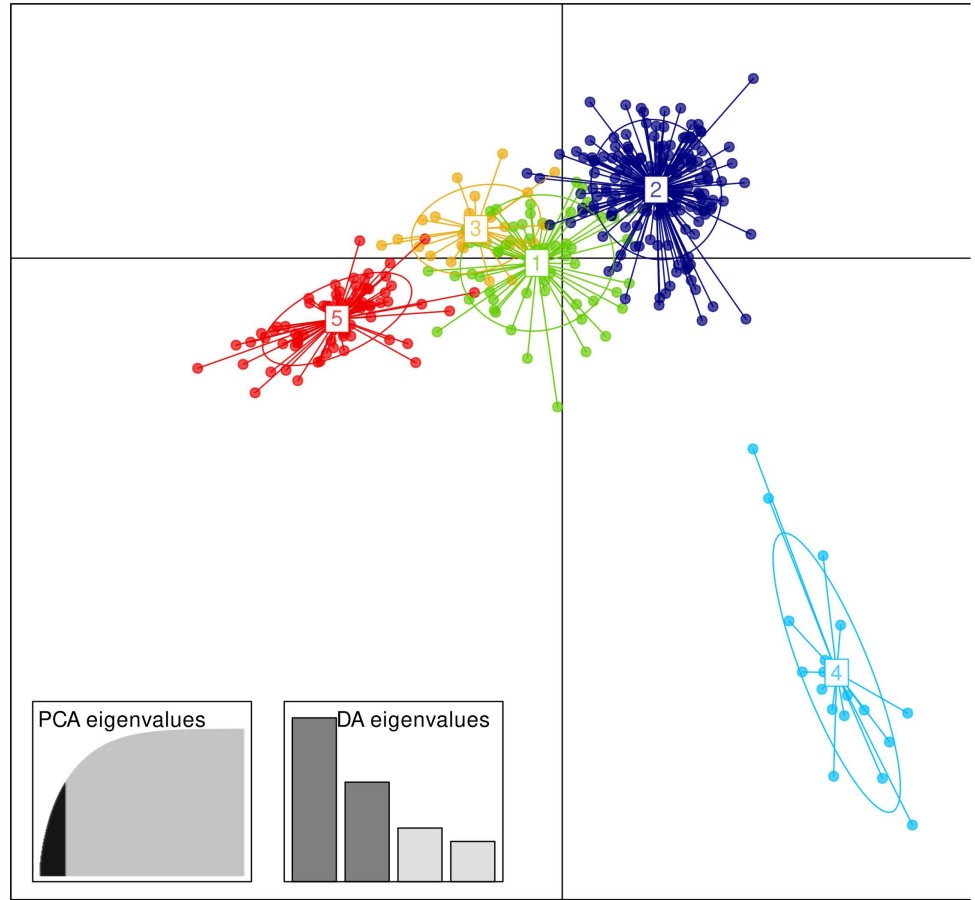

**Fig 8. DAPC plot showing five clusters.** Cluster 1: populations TLAH and HUAY from northern SMOr. Cluster 2: 10 populations from southern SMOr+TMVB (TENA, HUAC, HUAH, ZACA, TOTO, ACA, MINAS, ELO, TEPE, MIX). Cluster 3: 26 individuals from the VOVNI and MAD populations. Cluster 4: Most individuals of the TEPE population. Cluster 5: four northern populations (VOVNI, LT, MAD, and PAMO).

increasingly studied scenario in the Neotropics, where hybridization may provide selective advantages in highly heterogeneous habitats [41]. Further admixture analysis, including the sympatric species, would be required to test such hypothesis. Hybridization as a possible factor contributing to the high genetic diversity and divergence in these populations.

Genetic differentiation, measured as $F_{ST}$, is high (0.21) according to the intervals proposed by Hartl & Clarck [42]. However, rather than applying categorical thresholds from general frameworks, we interpret this magnitude in the ecological and geographical context of *T. sylvatica.* AMOVA showed that most genetic variation occurs within the populations (79.50%), with a smaller fraction among populations (11.84%). This pattern is consistent with other outcrossing and long-lived trees [43]. Direct data on *T. sylvatica* are lacking. However, entomophily (including bee pollination) has been reported in other *Ternstroemia* species such as *T. laevigata* and *T. dentata* [44], which would be expected to promote outcrossing and maintain high within-population genetic diversity while limiting the spatial scale of pollen movement. The fleshy, reddish seeds suggest bird dispersal [45], a syndrome that typically generates mainly local movement interspersed with occasional longer-distance events. The combination of short-range pollen flow and primarily local seed dispersal can maintain high within-population variation and produce measurable but not extreme among-population differentiation [46] across discontinuous areas—matching our results. For orientation only (not as interpretive anchors), published values for

other TMCF trees span a broad range; our estimate falls within the mid-range, and qualitative labels attached to similar values vary across taxa, markers, and sampling [9,10,13,39,43]. AMOVA showed that most genetic variation is partitioned (79.50%) whereas variation among populations is moderate (11.84%). These results are consistent with an outcrossing mating system with predominantly local pollen movement —likely mediated by small insects such as bees (like reported in *Ternstroemia laevigata* and *T. dentata* [44])—combined with restricted among-population dispersal across the mountainous landscape. The observed moderate differentiation between the northern and southern groups suggests a split between the gene pools since it explains a considerable amount of the genetic variation (8.67%).However, the substructure within the northern group is fully congruent with the distributional MB gap. Larger distributional gaps, such as the SGSP, separating the northernmost population of VOVNI from the rest of the north group (ca. 180 km), do not represent a significant barrier to genetic structure. Therefore, barriers other than the conspicuous SGSP and the MB gaps remain to be tested, specially within the southern group where STRUCTURE detected seven genetic groups.

### Genetic clustering

The clustering pattern obtained by STRUCTURE and DAPC is largely concordant when considering $K = 10$: STRUCTURE group 3 mostly corresponds to DAPC cluster 5, STRUCTURE group 7 corresponds to DAPC cluster 3, STRUCTURE group 10 corresponds to DAPC cluster 1, STRUCTURE groups 1, 5, 6, 8, 9, and 10 correspond to DAPC cluster 2, and STRUCTURE group 4 corresponds to the divergent DAPC cluster 4 (Most of TEPE individuals). Even the first-level STRUCTURE clustering did not segregate TEPE individuals. Populations from southern SMOr, TMVB, and the population from SMS largely overlap, according to DAPC. However, STRUCTURE revealed up to six genetic groups within these regions. Given the increasing fragmentation in eastern Mexico, this differentiation can be interpreted as accelerated genetic isolation [3,18]. However, testing such a hypothesis requires further gene flow and fragmentation analysis.

### Conclusions

The present study comprehensively assesses the spatial genetic diversity and population differentiation of *Ternstroemia sylvatica* across its distribution in eastern Mexico. Our findings reveal a significant genetic differentiation among populations, with most genetic variation occurring within populations rather than between them. The detected genetic structure suggests historical discontinuity and evidence of inbreeding, particularly in northern populations where a lower genetic diversity was observed. This pattern is consistent with previous studies on montane cloud forest species, highlighting the impact of biogeographic barriers such as the Moctezuma River basin in shaping genetic diversity and connectivity.

The identification of two major genetic groups, corresponding to northern and southern populations, suggests that *T. sylvatica* has undergone significant historical and ecological pressures, which have contributed to its current genetic landscape. The observed heterozygote deficit and high inbreeding coefficients in specific populations further underscore the importance of conservation efforts to mitigate genetic erosion. Specific populations with high genetic diversity may be influenced by introgression from sympatric *Ternstroemia* species, suggesting that hybridization may contribute to genetic variation.

The implications of this research extend to conservation strategies aimed at preserving genetic diversity within *T. sylvatica*. The pronounced genetic differentiation among populations suggests the need for regionally tailored conservation approaches, emphasizing the preservation of connectivity among populations to enhance genetic exchange. Future research should integrate ecological, demographic, and genomic data to further elucidate the adaptive potential of *T. sylvatica* in response to environmental changes and anthropogenic pressures within the rapidly changing landscapes of eastern Mexico.

### Supporting information

**S1 File. MLG accumulation curve, Hardy-Weinberg linkage disequilibrium test, and linkage disequilibrium tests.**
(DOCX)

## Acknowledgments

The authors thank Sol Cristians Niizawa, Luis Roberto Martínez Beiza, and Julio César Ramírez Martínez for their field support, Roberto Pedraza of Sierra Gorda A.C. for providing logistic support and access to areas within the Sierra Gorda Reserve, Leccinum José García Morales and José Rodrigo Carral Domínguez for providing pictures of *T. sylvatica*, and two anonymous referees whose very valuable and detailed comments significantly improved the impact and quality of this paper.

## Author contributions

**Conceptualization:** Hernán Alvarado-Sizzo.

**Data curation:** Hernán Alvarado-Sizzo, Isolda Luna-Vega, Edna Arévalo-Marín.

**Formal analysis:** Hernán Alvarado-Sizzo, Edna Arévalo-Marín.

**Funding acquisition:** Isolda Luna-Vega.

**Investigation:** Hernán Alvarado-Sizzo, Edna Arévalo-Marín.

**Methodology:** Hernán Alvarado-Sizzo, Edna Arévalo-Marín, Othón Alcántara-Ayala.

**Software:** Isolda Luna-Vega.

**Supervision:** Isolda Luna-Vega, Gerardo Rivas.

**Validation:** Gerardo Rivas.

**Writing – original draft:** Hernán Alvarado-Sizzo, Gerardo Rivas.

**Writing – review & editing:** Hernán Alvarado-Sizzo, Isolda Luna-Vega, Edna Arévalo-Marín, Othón Alcántara-Ayala, Gerardo Rivas.

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
