## [Decision Letter · Decision Letter 0]

5 Jun 2025

Dear Dr. Rivas,

Thank you for submitting your manuscript to PLOS ONE. After careful consideration, we feel that it has merit but does not fully meet PLOS ONE’s publication criteria as it currently stands. Therefore, we invite you to submit a revised version of the manuscript that addresses the points raised during the review process.

Your manuscript has been revised by two expert reviewers that consider that some issues need to be addressed. The introduction need to be modified to clearly state the aims of the study. Furthermore, reviewers also consider that there is some work to be done in the Discussion section. One of the reviewers found that some conclusions are not fully supported by your results. On the other hand, you should take into account the impact of sample size on some of the genetic variables such as the allelic richness (for instance, calculating the rarefied allelic richness). I would be delighted to reconsider your submission once you address all these comments.

We look forward to receiving your revised manuscript.

Kind regards,

Vicente Martínez López

Academic Editor

PLOS ONE

Journal Requirements:

5. We note that Figures 1,5 and 7 in your submission contain map/satellite images which may be copyrighted. All PLOS content is published under the Creative Commons Attribution License (CC BY 4.0), which means that the manuscript, images, and Supporting Information files will be freely available online, and any third party is permitted to access, download, copy, distribute, and use these materials in any way, even commercially, with proper attribution. For these reasons, we cannot publish previously copyrighted maps or satellite images created using proprietary data, such as Google software (Google Maps, Street View, and Earth). For more information, see our copyright guidelines: http://journals.plos.org/plosone/s/licenses-and-copyright.

 1. You may seek permission from the original copyright holder of Figures 1,5 and 7  to publish the content specifically under the CC BY 4.0 license. 

Reviewers' comments:

Reviewer's Responses to Questions

**Comments to the Author**

1. Is the manuscript technically sound, and do the data support the conclusions?

Reviewer #1: Partly

Reviewer #2: Yes

2. Has the statistical analysis been performed appropriately and rigorously?

Reviewer #1: No

Reviewer #2: Yes

3. Have the authors made all data underlying the findings in their manuscript fully available?

Reviewer #1: Yes

Reviewer #2: Yes

4. Is the manuscript presented in an intelligible fashion and written in standard English?

Reviewer #1: Yes

Reviewer #2: Yes

Reviewer #1: The study applied microsatellite for calculating genetic indices, nevertheless the calculation of Ar was without the consideration of sample size, which may cause bias. The conclusion based on comparing genetic diversity was made without statistical test. The language level of this manuscript was good.

Reviewer #2: Revision is attached as a WORD document.

Review of manuscript “Spatial genetic diversity and populational differentiation of Ternstroemia sylvatica (Ericales: Pentaphylacaceae) in eastern Mexico”

Introduction:

The first paragraph is directly describing the species, while to reach a broader audience it could be about the type of question that is going to be explored. For instance, it could be a paragraph about the statement in line 61 ‘[literature survey showing that] fragmentation affects the genetic diversity and future conservation of tree species’

Line 60: what does positive mean?

Line 61: Maybe spell out the first author’s name? e.g., Wehenkel et al. (2017) demonstrated…

Lines 66-67: a bit unclear question. Please try to re-phrase it.

Would commenting/advising about conservation plans for this species be also an aim?

Methods:

Is it known that the species is diploid? If no, how do you think this may affect the results?

Lines 71-72: add dates of collection. How many samples are collected per population? It is interesting that all material is fresh, and no herbarium material is used.

Line 73: grinded or ground

Line 76: S. sylvatica should be T. sylvatica

Line 97: please add the reference after each package name.

Line 104: why is K=1 not used?

Line 105: there’s an extra ‘e’ after the reference.

Line 109: please add reference after QGIS 3.22.4

Results:

I suggest adding a picture of the species.

Private alleles can be reported for each population.

Fst value between the northern and southern populations can be reported (based on genetic clustering).

Line 139, 143 and 151: pairwise Fst

Discussion:

Do these plants grow vegetatively too? Can it affect the genetic structure? And after how long of fragmentation, genetic effects would be observed?

Line 238: an extra ‘bitt’ after reference.

Line 241: add a space before ‘where’

Lines 242-243 gene flow instead of genetic flow

Lines 303-end: using private alleles at each population affects conservation action plans. You may consider it.

**Do you want your identity to be public for this peer review?** For information about this choice, including consent withdrawal, please see our Privacy Policy

Reviewer #1: No

Reviewer #2: No

---

## [Author Response · Author response to Decision Letter 1]

28 Jul 2025

We appreciate the feedback provided in the recent review of our manuscript. We appreciate the observations and suggestions from both you and the referees as they are positive, relevant, and constructive. Based on the review, we have integrated most suggestions, resulting in a new and improved version. All modifications are introduced using track changes as requested and attached to the resubmission, as well as the formatted final version. Additionally, we are attaching the observations regarding the Journal Requirements, as well as those made by the referees, along with our specific responses to each of them.

Journal Requirements Response:

R: We have carefully reviewed the PLOS ONE style requirements and, where necessary, applied the journal's guidelines.

R: We have removed funding information from the manuscript, keeping it only in the funding section of the submission form.

R: We have included our collection permission code issued by the Mexican

Government.

4. We note that the grant information you provided in the ‘Funding Information’ and ‘Financial

Disclosure’ sections do not match. When you resubmit, please ensure that you provide the correct grant numbers for the awards you received for your study in the ‘Funding Information’ section.

R: We have corrected grant information.

5. We note that Figures 1,5 and 7 in your submission contain map/satellite images which may be copyrighted. All PLOS content is published under the Creative Commons Attribution License (CC BY 4.0), which means that the manuscript, images, and Supporting Information files will be freely available online, and any third party is permitted to access, download, copy, distribute, and use these materials in any way, even commercially, with proper attribution. For these reasons, we cannot publish previously copyrighted maps or satellite images created using proprietary data, such as Google software (Google Maps, Street View, and Earth). For more information, see our copyright guidelines: http://journals.plos.org/plosone/s/licensesand-copyright.

R: We have replaced the Google Satellite layers in Figures 1,5, and 7 (figures 2, 3, and 7; respectively, in the new version) for a Digital Elevation Model under Open Data Policy, which is equivalent to CC BY 0 license (Public Domain). See terms of use in: https://en.www.inegi.org.mx/inegi/terminos.html. We have also included the reference for this layer.

Responses to Reviewer 1:

Introduction:

The first paragraph is directly describing the species, while to reach a wider audience it could be about the type of question that is going to be explored. For instance, it could be a paragraph about the statement in line 61 ‘[literature survey showing that] fragmentation affects the genetic diversity and future conservation of tree species’

R: We agree that reorganizing the information in the suggested way will improve the scope and the presentation of the context. We also agree that the suggested mention of fragmentation is pertinent.

Line 60: what does positive mean?

R: We changed the redaction to: these studies demonstrated a positive association between genetic diversity and population size. Thanks for your good suggestion.

Line 61: Maybe spell out the first author’s name? e.g., Wehenkel et al. (2017) demonstrated…

R: Done.

Lines 66-67: a bit unclear question. Please try to re-phrase it.

R: We rewrote this line for clarity, making the aims more explicit.

Would commenting/advising about conservation plans for this species be also an aim?

R: We consider that delving into conservation topics is far beyond the aims of this work. However, we are adding some mentions regarding the relevance of describing genetic parameters, such as diversity, rare alleles, and inbreeding index.

Methods:

Is it known that the species is diploid? If no, how do you think this may affect the results?

R: There are only cytogenetic studies on Ternstroemia japonica that report an haploid chromosome (Morinaga & Fukushima, 1931). There aren’t specific studies on T. sylvatica. Ploidy may be a significant issue for microsatellite analysis when individuals with different ploidy levels are present. However, most R packages and clustering

software like STRUCTURE are fully capable of analyzing polyploids. See the following references:

Kamvar, Z. N., Everhart, S. E., & Grünwald, N. J. Data preparation in Population genetics and genomics in R.

https://grunwaldlab.github.io/Population_Genetics_in_R/Data_Preparation.html

Pritchard, J. K., Wen, W., & Falush, D. (2003, July). Documentation for STRUCTURE software: Version 2.

https://citeseerx.ist.psu.edu/document?repid=rep1&type=pdf&doi=e05f0090ca1b0bdd9c198f13250af05548fc2db

0

Also, we found no evidence of polyploidy in the microsatellite electropherograms. This can be suspected when more than two alleles show up, but this was not the case (see reference below). Therefore, we are assuming T. sylvatica is a diploid. We added a mention of this.

Selkoe, K. A., & Toonen, R. J. (2006). Microsatellites for ecologists: a practical guide to using and evaluating

microsatellite markers. Ecology letters, 9(5), 615-629.

Lines 71-72: add dates of collection. How many samples are collected per population? It is interesting that all material is fresh, and no herbarium material is used.

R: We added the collection years. The sample size per population ranged from 15 to 30 individuals, as shown in Table 2. However, considering that this information is relevant to the sampling description, we are repeating it in Table 1. In previous molecular work with the genus, we have encountered numerous problems using herbarium-preserved specimens. These samples are used to extract a dark precipitate (presumably a mixture of tannins) that hinders DNA isolation and decreases the efficiency of PCR. Additionally, newspaper-preserved specimens consistently result in DNA degradation, leading to a decrease in PCR success.

Line 73: grinded or ground

R: Correction done.

Line 76: S. sylvatica should be T. sylvatica

R: Correction done.

Line 97: please add the reference after each package name.

R: Citations added.

Line 104: why is K=1 not used?

R: The method proposed by Evanno et al. (2005) is based on the second-order rate of change (second derivative) of the log probability of the data, LnP(D), between successive K values (number of clusters). So, since K=1 has no previous values (there are no values for K=0), ΔK cannot be calculated.

Line 105: there’s an extra ‘e’ after the reference.

R: Correction done.

Line 109: please add reference after QGIS 3.22.4

R: SIG reference added.

Results:

I suggest adding a picture of the species.

R: We consider this to be quite pertinent and illustrative, so we have included a plate of T. sylvatica as Fig 1.

Private alleles can be reported for each population.

R: We have included the calculation of private alleles.

Fst value between the northern and southern populations can be reported (based on genetic clustering).

R: This is reported in the “Between region” comparison as FCT. We have made this explicit.

Line 139, 143 and 151: pairwise Fst

R: Correction done.

Discussion:

Do these plants grow vegetatively too? Can it affect the genetic structure? And after how long of fragmentation, genetic effects would be observed?

R: We did not find reports about vegetative reproduction of Mexican Ternstroemia. Certainly, the presence of clones can artificially modify the genetic structure, but we were cautious to perform a Multilocus Genotype test that (See supplementary information S1) demonstrated the absence of clonal individuals. Finally, demonstrating the effects of fragmentation on genetic diversity or structure requires at least two generations or cohorts after the event, allowing for the calculation of genetic parameters over two generations. In this case, however, we collected only the reproductive individuals, limiting any temporal/generational inference.

Line 238: an extra ‘bitt’ after reference.

R: Typing error, corrected.

Line 241: add a space before ‘where’

R: Typing error, corrected.

Lines 242-243 gene flow instead of genetic flow

R: Correction done.

Lines 303-end: using private alleles at each population affects conservation action plans. You may consider it.

R: We included the private alleles calculations.

Responses to Reviewer 2:

This study investigated the spatial patterns of population genetic diversity and differentiation of a typical cloud forest tree species Ternstroemia sylvatica in eastern Mexico. Using microsatellite markers, the authors estimated genetic indices and applied STRUCTURE and DAPC analysis to identify different genetic clusters. These clusters appear to be explained by

geographic separation and local hybridization. Before this study can be published for a broader audience, I have the following thoughts that the authors should consider.

Major comments:

1) Introduction:

The goal of this study was not very clearly stated in the introduction. If the primary aim was to present the distribution of population genetic diversity and differentiation of the species, the broader relevance and appeal to a wider scientific audience may be limited. The author should more explicitly articulate the state of the art the study addresses. If the study aims to test the hypothesis that geographical patterns have shaped the genetic structure, e.g. SGSP and MB, this hypothesis should be clearly formulated in the beginning. Additionally, fragmentation is mentioned both in the introduction and later, yet the definition is unclear. What is the difference between fragmentation and biogeographical barrier in the manuscript?

R: We agree that the use of the term “fragmentation” (when referring to anthropogenically driven fragmentation) may have been misleading or even confused with the “biogeographical discontinuities”. We are explicitly integrating the biogeographic hypothesis. Specifically, we are testing the role of the major disjunctions within the distribution. Also, we have summarized the

leading causes and effects of human-driven fragmentation as context or justification for this work.

2) Methodology:

the AMOVA was not stated in the Methods section. Moreover, the terminology (region, group?) and the interpreted based on this need to be more carefully addressed through out the manuscript.

R: We have included the AMOVA method, along with the methodological details used to conduct it. Additionally, we have standardized the use of hierarchical terms to avoid confusion, especially when it comes to variance sources like

regions, populations, and individuals; instead of the generic and often misleading term “samples” generated in the package poppr by default.

3) Interpretation of the results:

When interpreting the results, the manuscript would benefit from including statistical analysis and adding references or comparisons with other studies or species. If no studies were found, theoretical work should provide a frame. This would help to substantiate the claims made based on the observed values (see below for specific points). Several studies were briefly mentioned in the introduction, however without relationship to the result of the current study.

R: We revised the Discussion section to clearly explain what the statistical results mean in terms of population structure and genetic differentiation. Although these analyses (e.g., Hardy-Weinberg tests, AMOVA/F-statistics, DAPC, STRUCTURE) were already presented in detail in the Results section, we now highlight their relevance more clearly by interpreting their implications for population structure and genetic differentiation. We also draw comparisons with findings from previous studies to substantiate our conclusions and provide a stronger contextual framework for our results. In parallel, we added new references and re-evaluated those already mentioned in the Introduction, ensuring they are meaningfully integrated into the

Discussion. These additions help position our results within the broader literature and enhance the clarity and impact of the study's contributions. We believe these revisions significantly improve the manuscript and address the reviewer’s thoughtful suggestions.

4) Discussion structure:

The structure of the discussion should be improved. Under the first subtitle, the content

is not well structured. While the second and last paragraph contain some repetition to

the content in the first subtitle and partly disconnected.

R: We have generated a proposal to restructure the discussion so that it is

clearer and reduces repetition. This allows the information to be read from a

general perspective to a specific one.

5) Conclusion:

After reading the manuscript, I find the conclusion that that species exhibits generally general high genetic differentiation is not fully supported. Most of the genetic variance is found within population not between them. The interpretation of inbreeding

depression should be first properly addressed in discussion by comparison with other studies. The last paragraph regarding conservation implementation feels disconnected and relies heavily on general statements without sufficient support from the findings. I recommend that the authors focus more closely on summarizing the main results and their direct implications, and avoid broader claims that are not firmly grounded in the data.

R: We appreciate the reviewer’s comments on the conclusion and acknowledge the importance of ensuring that our interpretations and recommendations are firmly grounded in the data. Below, we addressed each point in detail:

1. On genetic differentiation:

We respectfully disagree with the notion that high genetic differentiation is not supported. While it is true that most genetic variation resides within populations (see references below in point 2)—a common pattern in outcrossing, long-lived plant species, like seems to be the case in Ternstroemia—the observed FST values in our study are within the range typically interpreted as moderate to high genetic differentiation (e.g., Wright 1978; Hartl & Clark 1997). To strengthen this conclusion, we have now included additional examples from the literature reporting similar F ST values in comparable species, reinforcing the biological significance of our findings.

2. On within-population variation and broader patterns:

We fully agree that forest tree species, particularly those in temperate ecosystems, often show high within-population genetic diversity. We have clarified this in the Discussion and contextualized our results by referencing foundational studies such as:

Loveless, M. D., & Hamrick, J. L. (1984). Ecological determinants of genetic structure in plant populations. Annual review of ecology and systematics, 65-95.

Hamrick, J. (1989). The genetic structure of tropical tree population: Associations with reproductive biology. The Evolutionary Ecology of Plants, 129-146.

Hamrick, J. L., Godt, M. J. W., & Murawski, D. A. and Loveless, M.D. (1991). Genetics and conservation of rare plants, 75.

The above cited references describe how life history traits and mating systems, shape genetic structure in plants. These references help reconcile our finding of substantial within-population variance with the evidence of overall differentiation between populations.

3. On inbreeding depression:

We have revised the Discussion to better substantiate our interpretation regarding inbreeding depression. We now reference studies that have documented inbreeding effects in forest tree specie

---

## [Decision Letter · Decision Letter 1]

15 Sep 2025

Spatial genetic diversity and populational differentiation of Ternstroemia sylvatica (Ericales: Pentaphylacaceae) in eastern Mexico

PLOS ONE

Dear Dr. Rivas,

Thank you for submitting your manuscript to PLOS ONE. After careful consideration, we feel that it has merit but does not fully meet PLOS ONE’s publication criteria as it currently stands. Therefore, we invite you to submit a revised version of the manuscript that addresses the points raised during the review process.

Reviewers have recognized a significant improvement in the clarity and quality of the manuscript. However, they have also identified some issues that require your attention. Please, pay particular attention to the comments on the discussion section to ensure that the terms used are adequate in the context of your research.

We look forward to receiving your revised manuscript.

Kind regards,

Vicente Martínez López

Academic Editor

PLOS ONE

Journal Requirements:

Additional Editor Comments:

Reviewer #1:

Reviewer #2:

Reviewers' comments:

Reviewer's Responses to Questions

**Comments to the Author**

Reviewer #1: (No Response)

Reviewer #2: All comments have been addressed

2. Is the manuscript technically sound, and do the data support the conclusions?

Reviewer #1: (No Response)

Reviewer #2: Yes

3. Has the statistical analysis been performed appropriately and rigorously?

Reviewer #1: (No Response)

Reviewer #2: Yes

4. Have the authors made all data underlying the findings in their manuscript fully available?

Reviewer #1: (No Response)

Reviewer #2: Yes

5. Is the manuscript presented in an intelligible fashion and written in standard English?

Reviewer #1: (No Response)

Reviewer #2: Yes

Reviewer #1: General comments: The manuscript has undergone substantial improvement and the overall message and aims of the study are now clearer. However, before it can be considered for publication, the main concern remains the first aim: evaluating genetic diversity and differentiation. I strongly encourage the authors to remain closely aligned with their data and to use appropriately comparable studies to support their statements. Please avoid overstatements and ensure precise and careful use of terms such as “genetic erosion”, “inbreeding depression”.

The structure of the discussion has improved considerably, but I recommend further effort to reconsider the placement of some content, organizing it around the main messages rather than the workflow, while also avoiding repetition.

Please also adjust the abstract accordingly.

Specific points are detailed below.

Line 49: The region of Sierra Gorda may need some explanation. I suggest providing a map (or adding to the existing map) of the relevant geographical and/or administrative units and adding this information in the Table 1 for each sampled population. It would greatly improve readability for readers unfamiliar with the region, particularly in the later sections when the relationship between population genetic pattern and geographical pattern is discussed.

Line 56: The meaning of “have not been evaluated” is unclear. Please clarify what the author intends to convey here? Dose this imply that the aspect is highly relevant but has not yet been assessed, or that it is not relevant in this context.

Lines 65–70: This sentence is overly long and dense, making it difficult to follow. I suggest breaking it into two or more sentences and clarifying the main points. In addition, the broad generalization (“have demonstrated isolation by distance…, low to moderate genetic diversity…”) may be problematic, as not all of the cited studies necessarily report all of these patterns. Please ensure that the summary accurately reflects the findings of each reference.

Line 74: “resulting” should be “resulted”

Line 75: The phrase “land-change use to …” is unclear. Suggest revising to “land-use change due to agriculture and cattle raising” for clarity.

Line 81-82: “Moreover, the genus….of its genetic variation” is repetitive with the Line 79-80: “the lack of knowledge… genetic diversity”. Suggest merging or removing.

Table 1: The Acronym “Tepe” should be “TEPE”.

Line 129: The settings of the DAPC should be reported in detail. Please specify the discrimination functions and the number of principal components used.

Line 132: The methods state that the definitions of “northern group” and “southern group” are based on the first-level STRUCUTRE analysis, but the details of which populations are included are provided at the end of the result. This makes the results difficult to follow. Suggest adding this information already here or at least proving a clear reference to where it is given.

Line 142: “Poliploidy” should be “polyploidy”.

Line 163: “higher” should be “lower”. In addition, why there are no comparisons of Ho and Ar (AR)? If both measures have been reported and the intention is to compare genetic diversity between groups, or state that the genetic diversity is generally lower in the north than in the south, why is only He considered? Suggest either omitting Ar and Ho or state the rationale for focusing on only one measure.

Figure 2, 3 and 4: Acronym “OVNI” should be “VOVNI”

Line 201: “hereafter referred to as the northern group” is incorrect as “northern group” was already used before in Methods and Results. Suggest change to “referred to as “northern group” throughout the study”. The same applies to “southern group”. See also comment at Line 132.

Line 225: “MAD” should be “PAMO” I assume.

Line 241: Please clarify why two numbers are reported when only one species is considered—are these values derived from different studies? In addition, the corresponding reference is missing and should be provided.

Line 241-Line 243: For readability, consider combining into a single sentence. For example: “It is comparable to T. lineate (He=…), which is similar in ecological terms (temperate forest dwellers), has similar fragmentation issues and was evaluated using the same microsatellite set.” However, note that T. sylvatica was stated as a tropical instead of temperate species (Line 41), so this statement may need revision.

Line 245: Please standardize the reporting of values to two decimal places throughout the manuscript.

Line 249: The reported He of A. guatemalensis is below 0.1, whereas the He of Magnolia species was interpreted as media to high. In pine, He is mostly related to population size but not isolation. Therefore, the comparison with the cited studies may not be appropriate. Consequently, the conclusion that T. sylvatica suffers from genetic erosion is not fully supported and should be revised. Also, the explanation the low diversity using attribution to population isolation at this place is arbitrary.

Nevertheless, the small population size in the current study, only about 20 individuals per population and all available adults were sampled, could be a good explanation for the low He, if the author insists to interpret as such.

Line 250-260: The MB gap falls within the region of “northern group”. If the authors wish to state this subdivision, other clearly defined terms other than “northern, low-diversity group” and “southern, high-diversity group” should be used as this conflict with the defined terms in Line 201-202. One idea could be to report this subdivision in the results (Line 222-Line 228).

Hidalgo should be at least shown on the map so that the referred populations are clear (see also the comment at Line 49). In addition, the content here seems misplaced. I suggest moving this section together with Lines 293-307, as both address this biogeographical pattern.

Line 264: “notorious” change to “noteworthy”

Line 264: Specify which measure was considered for this statement “high diversity spot”.

Line 270-275: I cannot agree with the interpretation of the author. Although Wright (1978) and Hartl & Clark (1997) were cited, these works provide only a general framework. The interpretation of values in empirical studies requires contextual discussion. Moreover, the interpretations in the referred studies do not support the authors’ conclusions. For example, Magnolia species with a global FST of 0.21 and pairwise FST values of 0.03–0.35 were interpreted as showing low to moderate differentiation. In Oreopanax, genetic structure was found among age clusters, but no interpretation of high genetic differentiation was made. Similarly, Fi values of M. schiedeana (0.21–0.28) were interpreted as moderate. Drawing conclusions based solely on arbitrary thresholds, without linking them to the ecological and geographical context, is misleading.

Line 275-283: The interpretation that “genetic variation being mostly within populations indicates outcrossing through local, within-population gene flow realized by small insects such as bees” is not logical to me. If gene flow occurred predominantly within populations, one would expect greater differentiation among populations. This point requires clarification.

Line 282: Add “within population” before “genetic differentiation”. It is otherwise counterintuitive why long-distance seed dispersal by birds would increase overall genetic differentiation.

Line 316-320: This content could be merged with Line 265-269

Line 321: The term “metapopulation” is misused in this context. There is no evidence of metapopulation dynamics, and DAPC does not provide information on processes such as local extinction or colonization. I suggest removing this sentence.

Line 330-332: The use of the term “fragmentation” is unclear, as no analysis of landscape fragmentation was conducted. The authors may wish to specify the intended meaning to avoid misunderstanding. In addition, the interpretation of “limited gene flow” and “inbreeding depression” is not supported by the results. Inbreeding depression is a consequence of inbreeding; based on the data, the evidence supports a certain degree of inbreeding but not inbreeding depression. I suggest revising this sentence to: “limited gene flow between northern and southern distribution areas” and “show a certain degree of inbreeding.”

Reviewer #2: Dear Authors,

Thank you for your great work. All my comments have been addressed except one; to add a paragraph to the Introduction, describing scope of the study and the gap of knowledge. Otherwise, the MS has a good quality.

**Do you want your identity to be public for this peer review?** For information about this choice, including consent withdrawal, please see our Privacy Policy

Reviewer #1: No

Reviewer #2: No

---

## [Author Response · Author response to Decision Letter 2]

22 Oct 2025

Reviewer #1: General comments: The manuscript has undergone substantial improvement and the overall

message and aims of the study are now clearer. However, before it can be considered for publication, the main

concern remains the first aim: evaluating genetic diversity and differentiation. I strongly encourage the authors to

remain closely aligned with their data and to use appropriately comparable studies to support their statements.

Please avoid overstatements and ensure precise and careful use of terms such as “genetic erosion”, “inbreeding

depression”.

The structure of the discussion has improved considerably, but I recommend further effort to reconsider the

placement of some content, organizing it around the main messages rather than the workflow, while also

avoiding repetition.

Please also adjust the abstract accordingly.

Specific points are detailed below.

R. We appreciate the reviewer’s positive feedback and take the remaining concerns seriously. In the

revision, we (1) strictly align Aim 1 with our own estimates (e.g., FST, AMOVA) and remove categorical

labels from generic thresholds; any cross-taxon numbers are used only as numerical orientation with

explicit caveats; (2) remove overstatements and tighten terminology throughout—replacing “genetic

erosion” and “inbreeding depression” with precise, data-supported language; (3) limit comparisons to

studies that are methodologically comparable and state limitations where they differ; (4) restructure the

Discussion around the main messages (pattern of diversity; spatial structure/biogeographic break;

conservation implications), relocating or removing repetitive content; and (5) adjust the Abstract

according to these changes. We also improved readability by adding subregional context to the map and

Table 1, standardizing reporting, and clarifying methods where requested.

Line 49: The region of Sierra Gorda may need some explanation. I suggest providing a map (or adding to the

existing map) of the relevant geographical and/or administrative units and adding this information in the Table 1

for each sampled population. It would greatly improve readability for readers unfamiliar with the region,

particularly in the later sections when the relationship between population genetic pattern and geographical

pattern is discussed.

R: We appreciate the thoughtful suggestion from the reviewer. In fact, subregionalization is an aspect we

overlooked because of our familiarity with the area. We have updated Figure 2 to display the referenced

subregions and added regionalization and subregionalization to Table 1. We believe this information can

offer a better spatial context for potential readers.

Line 56: The meaning of “have not been evaluated” is unclear. Please clarify what the author intends to convey

here? Dose this imply that the aspect is highly relevant but has not yet been assessed, or that it is not relevant in

this context.

R: Certainly, as the reviewer pointed out, this sentence was unclear: we aimed to highlight the lack of

assessment of the effects of harvesting on T. sylvatica as a knowledge gap. We have added a clear

sentence indicating the absence of studies on species sustainability related to extraction.

Lines 65–70: This sentence is overly long and dense, making it difficult to follow. I suggest breaking it into two or

more sentences and clarifying the main points. In addition, the broad generalization (“have demonstrated

isolation by distance..., low to moderate genetic diversity...”) may be problematic, as not all of the cited studies

necessarily report all of these patterns. Please ensure that the summary accurately reflects the findings of each

reference.

R: Thank you for this constructive comment. We agree that the sentence was too long and that our

wording could be interpreted as a broad generalization. We have split the section into shorter sentences

and softened the claims so that the appropriate subset of studies supports each point. In the revision,

we state that regional spatial genetic structure and limited gene flow are commonly reported (e.g., 5,7–

9,12), and that several studies examining disturbance/demography report a positive association between

population size and genetic diversity (11,13,14). We believe this revised text more accurately reflects the

findings of the cited literature.

Line 74: “resulting” should be “resulted”

R: We have applied this correction.

Line 75: The phrase “land-change use to ...” is unclear. Suggest revising to “land-use change due to agriculture

and cattle raising” for clarity.

R: We have applied for this correction.

Line 81-82: “Moreover, the genus....of its genetic variation” is repetitive with the Line 79-80: “the lack of

knowledge... genetic diversity”. Suggest merging or removing.

R: We agree that the sentence is repetitive; therefore, we removed it.

Table 1: The Acronym “Tepe” should be “TEPE”.

R: We have corrected the acronym to use uppercase letters.

Line 129: The settings of the DAPC should be reported in detail. Please specify the discrimination functions and

the number of principal components used.

Thank you for the suggestion. We have added the DAPC settings to the Methods (lines 1137–1147 now):

we retained 29 principal components (selected by alpha-score optimization) and 4 discriminant

functions (K = 5 groups; therefore K − 1 = 4).

Line 132: The methods state that the definitions of “northern group” and “southern group” are based on the first-

level STRUCUTRE analysis, but the details of which populations are included are provided at the end of the

result. This makes the results difficult to follow. Suggest adding this information already here or at least proving a

clear reference to where it is given.

R: We agree that the lack of context regarding the correspondence between populations and the first-

level genetic groups might confuse the readers. Therefore, we have updated this section to include the population

IDs of both groups.

Line 142: “Poliploidy” should be “polyploidy”.

R: We made this correction.

Line 163: “higher” should be “lower”. In addition, why there are no comparisons of Ho and Ar (AR)? If both

measures have been reported and the intention is to compare genetic diversity between groups, or state that the

genetic diversity is generally lower in the north than in the south, why is only He considered? Suggest either

omitting Ar and Ho or state the rationale

R: We have replaced 'lower' with 'higher' because it was mistakenly stated. We considered Expected

heterozygosity (He) since it is generally regarded as a more robust and informative measure of genetic

diversity than observed heterozygosity (Ho) or allelic richness (Ar). This is due to its statistical

properties and biological interpretation: it is less biased by sample size, incorporates allele frequency

distributions, and provides a comparable and biologically meaningful estimate of genetic diversity.

Instead of including the previous explanation, it is sufficient to focus on the comparisons of He as

suggested.

Figure 2, 3 and 4: Acronym “OVNI” should be “VOVNI”

R: This acronym has been corrected in all the figures.

Line 201: “hereafter referred to as the northern group” is incorrect as “northern group” was already used before

in Methods and Results. Suggest change to “referred to as “northern group” throughout the study”. The same

applies to “southern group”. See also comment at Line 132.

R: As the reviewer rightly points out, we already defined these groups in the methods section, so we

agree that it is unnecessary to specify "hereafter" and we have applied the suggestion.

Line 225: “MAD” should be “PAMO” I assume.

R: We reviewed the individual DAPC assignment, and it is correct: cluster number 3 (yellow) contains

individuals from VOVNI and MAD, not PAMO. However, the rest of MAD individuals belong to cluster

number 5 (blue), hence the duplication of the MAD label among the clusters.

Line 241: Please clarify why two numbers are reported when only one species is considered—are these values

derived from different studies? In addition, the corresponding reference is missing and should be provided.

R: Early versions of the manuscript cited values from a published work on T. lineata and from

unpublished research on T. huasteca. However, according to PLOS ONE policies, unpublished data

requires including the entire manuscript as a supplementary file. We considered that this would

complicate things just to support a single value, so we decided to omit it. However, we forgot to remove

the T. huasteca value. We left only the T. lineata published value.

Line 241-Line 243: For readability, consider combining into a single sentence. For example: “It is comparable to

T. lineate (He=...), which is similar in ecological terms (temperate forest dwellers), has similar fragmentation

issues and was evaluated using the same microsatellite set.” However, note that T. sylvatica was stated as a

tropical instead of temperate species (Line 41), so this statement may need revision.

R: We appreciate the reviewer’s suggestion and have made changes to improve readability. Additionally,

we must admit that although T. sylvatica is found in Tropical Montane Cloud Forest, it is not exclusive to this

vegetation type and also occurs in some temperate forest ecosystems. Therefore, we have updated

this in the abstract and Introduction.

Line 245: Please standardize the reporting of values to two decimal places throughout the manuscript.

R: We rounded all decimal numbers throughout the manuscript. However, the X and Y coordinates in

Table 1 were not rounded, because two decimal places do not help locate populations on a map.

Conversely, the populations projected on the map in Fig. 2 could potentially eliminate the need to keep X

and Y data in Table 1.

Line 249: The reported He of A. guatemalensis is below 0.1, whereas the He of Magnolia species was

interpreted as media to high. In pine, He is mostly related to population size but not isolation. Therefore, the

comparison with the cited studies may not be appropriate. Consequently, the conclusion that T. sylvatica suffers

from genetic erosion is not fully supported and should be revised. Also, the explanation the low diversity using

attribution to population isolation at this place is arbitrary.

Nevertheless, the small population size in the current study, only about 20 individuals per population and all

available adults were sampled, could be a good explanation for the low He, if the author insists to interpret as

such.

R: We acknowledge this point and agree that cross-species comparisons of He can be problematic due

to differences in demographic history, mating systems, marker sets, and analytical criteria. Therefore,

we removed those comparisons as direct evidence and reframed our interpretation to emphasize the

intraspecific pattern.

Line 250-260: The MB gap falls within the region of “northern group”. If the authors wish to state this subdivision,

other clearly defined terms other than “northern, low-diversity group” and “southern, high-diversity group” should

be used as this conflict with the defined terms in Line 201-202. One idea could be to report this subdivision in the

results (Line 222-Line 228).

R: Thank you for this helpful suggestion. We agree that our use of “northern, low-diversity group” and

“southern, high-diversity group” was ambiguous. We have revised the text to separately reference the

biogeographic disjunction of the Moctezuma Basin (MB) and the “northern/southern genetic groups”

defined by the first-level genetic clustering. Following your recommendation, we described the

“northern group” subdivision in relation to the manuscript, reporting the He values associated with each

side of the MB. In the revised text, we specify that populations north of the MB include VOVNI, LT, MAD,

and PAMO (with He = 0.53–0.60), while populations south of the MB show moderate to high diversity (up

to He = 0.73). We explicitly link these patterns to Figure 3 (interpolated He) and note that the

distributional break across the MB shown in Figure 2 corresponds to the transition in genetic diversity.

Finally, to avoid introducing new terminology, we removed the terms “low-diversity group” and “high-

diversity group”. We hope this resolves the inconsistency and improves clarity. Hidalgo should be at least

shown on the map so that the referred populations are clear (see also the comment at Line 49). In addition,

the content here seems misplaced. I suggest moving this section together with Lines 293- 307, as both

address this biogeographical pattern.

R: Since Hidalgo is a political boundary, it is irrelevant to biogeographical discussion. Therefore, we

replaced the Hidalgo region we intended to refer to with the term "southern SMOr," which we already

introduced earlier. However, in biogeographical terms, the coincidence with the quote above is

sufficiently relevant to keep this section.

Line 264: “notorious” change to “noteworthy”

R: We made the change.

Line 264: Specify which measure was considered for this statement “high diversity spot”.

R: We are now specifying that this refers to He.

Line 270-275: I cannot agree with the interpretation of the author. Although Wright (1978) and Hartl & Clark

(1997) were cited, these works provide only a general framework. The interpretation of values in empirical

studies requires contextual discussion. Moreover, the interpretations in the referred studies do not support the

authors’ conclusions. For example, Magnolia species with a global FST of 0.21 and pairwise FST values of

0.03–0.35 were interpreted as showing low to moderate differentiation. In Oreopanax, genetic structure was

found among age clusters, but no interpretation of high genetic differentiation was made. Similarly, Fi values of

M. schiedeana (0.21–0.28) were interpreted as moderate. Drawing conclusions based solely on arbitrary

thresholds, without linking them to the ecological and geographical context, is misleading.

R: We agree and have revised the text accordingly. We now interpret the magnitude of FST=0.21 in the

specific ecological and geographical context of T. sylvatica. Where we retain numbers from other TMCF

trees, they are used purely as orientation (not as interpretive anchors), acknowledging that qualitative

labels vary across taxa, markers, and sampling designs. We emphasize our own AMOVA results (79.5%

within; 11.84% among populations) and frame the pattern as moderate spatial structure consistent with

an outcrossing tree in discontinuous montane forest. We also corrected language to avoid drawing

conclusions on mechanisms that our data does not support.

We appreciate this clarification and agree that our wording conflated local mating with inter-population

gene flow. In the revision, we (1) replace “within-population gene flow” with “short-distance pollen

dispersal (local mating),” (2) note explicitly that gene flow in the context of FST refers to exchange among

populations, and (3) explain that the combination of local pollen movement and restricted inter-

population connectivity across rugged topography can produce high within-population diversity together

with non-trivial (but not extreme) among-population differentiation, in line with FST=0.21 and our AMOVA.

We further acknowledge that the observed differentiation likely reflects a balance between limited

contemporary movement and occasional inter-population connectivity (e.g., rare bird-mediated seed

dispersal) and/or historical connectivity, which can keep FST from becoming very high. We removed

the incorrect phrase “increases individual genetic differentiation within populations.”

Line 275-283: The interpretation that “genetic variation being mostly within populations indicates outcrossing

through local, within-population gene flow realized by small insects such as bees” is not logical to me. If gene

flow occurred predominantly within populations, one would expect greater differentiation among populations.

This point requires clarification.

R: Thank you for pointing this out. We agree that our wording

---

## [Decision Letter · Decision Letter 2]

25 Nov 2025

Dear Dr. Rivas,

Thank you for submitting your manuscript to PLOS ONE. After careful consideration, we feel that it has merit but does not fully meet PLOS ONE’s publication criteria as it currently stands. Therefore, we invite you to submit a revised version of the manuscript that addresses the points raised during the review process.

Thank you for your thorough revision of the manuscript. I am pleased to see the significant improvements achieved. Before the manuscript can be accepted, there are still a few minor issues that need to be addressed. Please submit the necessary corrections at your earliest convenience, and I will do my best to proceed with a smooth final decision.

We look forward to receiving your revised manuscript.

Kind regards,

Vicente Martínez López

Academic Editor

PLOS ONE

Journal Requirements:

Reviewers' comments:

Reviewer's Responses to Questions

**Comments to the Author**

Reviewer #1: All comments have been addressed

2. Is the manuscript technically sound, and do the data support the conclusions?

Reviewer #1: Yes

3. Has the statistical analysis been performed appropriately and rigorously?

Reviewer #1: Yes

4. Have the authors made all data underlying the findings in their manuscript fully available?

Reviewer #1: Yes

5. Is the manuscript presented in an intelligible fashion and written in standard English?

Reviewer #1: Yes

Reviewer #1: The manuscript has been substantially improved in both clarity and overall quality. I have the following minor issues that should be addressed.

Since the revised version does not include line numbers, I have added them, starting from the Abstract (excluding empty lines and tables) so it can be more easily followed.

Line 71: mark also the SGSP gap and MB gap in the map (Figure 2)

Line 206, Line 239: missing “)”.

Line 224-226, Line 263-265, Lines 280-283: these sentences do not convey any meaningful information. Please revise or consider delete.

Line 232: if the sampling scheme has already been exhausted, I doubt that suggesting a denser sampling makes sense. Please revise.

Line 267: name the tree species

Lines284-288: is this part repetitive to the content from lines 265-269? Also, name the species in the citation 44. If there are not repetitive, consider combine these two parts together.

Line 288: delete “within-population”.

**Do you want your identity to be public for this peer review?** For information about this choice, including consent withdrawal, please see our Privacy Policy

Reviewer #1: No

---

## [Author Response · Author response to Decision Letter 3]

28 Nov 2025

Reviewer #1: The manuscript has been substantially improved in both clarity and overall quality. I have the following minor issues that should be addressed.

Since the revised version does not include line numbers, I have added them, starting from the Abstract

(excluding empty lines and tables) so it can be more easily followed.

R: We thank the reviewer for the detailed review that has helped improve the MS. We

especially acknowledge the follow-up the reviewer has done throughout the review of

the work.

Line 71: mark also the SGSP gap and MB gap in the map (Figure 2)

R: We have included the SGSP and MB gaps labels and bars as they were in previous versions.

Line 206, Line 239: missing “)”.

R: We have applied this change

Line 224-226, Line 263-265, Lines 280-283: these sentences do not convey any meaningful

information. Please revise or consider delete.

R: L. 224-226: We agree that the wording was vague, so we modified it to highlight the difficulty

in discussing our diversity estimations, given the nature of the other studied species. Finally,

we redirected this section to the discussion of intrapopulation genetic variation.

L. 263-265: Yes, it might be risky to support that claim without a broader discussion, so we will

omit it.

L. 280-283: We agree that this section is superfluous, so we are removing it.

Line 232: if the sampling scheme has already been exhausted, I doubt that suggesting a denser

sampling makes sense. Please revise.

R: We agree. We removed the reference to the sampling scheme.

Line 267: name the tree species

R: We have included the names.

Lines 284-288: is this part repetitive to the content from lines 265-269? Also, name the species in the

citation 44. If there are not repetitive, consider combine these two parts together.

R: Yes, we removed the contents of L. 265-269 and retained the wording that is directly related

to the AMOVA results.

Line 288: delete “within-population”.

R: We have applied this change.

---

## [Editor Report · Decision Letter 3]

1 Dec 2025

Spatial genetic diversity and populational differentiation of Ternstroemia sylvatica (Ericales: Pentaphylacaceae) in eastern Mexico

PONE-D-25-08656R3

Dear Dr. Rivas,

We’re pleased to inform you that your manuscript has been judged scientifically suitable for publication and will be formally accepted for publication once it meets all outstanding technical requirements.

Please, revise the parenthesis on line 255, I think that it should be (He= 0.68).

Kind regards,

Vicente Martínez López

Academic Editor

PLOS ONE
---

## [Editor Report · Acceptance letter]

PONE-D-25-08656R3

PLOS One

Dear Dr. Rivas,

I'm pleased to inform you that your manuscript has been deemed suitable for publication in PLOS One. Congratulations! Your manuscript is now being handed over to our production team.

Kind regards,

on behalf of

Dr. Vicente Martínez López

Academic Editor

PLOS One